# Assessing the balance between excitation and inhibition in chronic pain through the aperiodic component of EEG

Cristina Gil Avila[1,2], Elisabeth S May[1,2], Felix S Bott[1,2], Laura Tiemann[1,2], Vanessa Hohn[1,2], Henrik Heitmann[1,2,3,4], Paul Theo Zebhauser[1,2,3], Joachim Gross[5], Markus Ploner[1,2,3]*

[1]Department of Neurology, TUM School of Medicine and Health, Technical University of Munich (TUM), Munich, Germany; [2]TUM-Neuroimaging Center, TUM School of Medicine and Health, TUM, Munich, Germany; [3]Center for Interdisciplinary Pain Medicine, TUM School of Medicine and Health, TUM, Munich, Germany; [4]Department of Psychosomatic Medicine and Psychotherapy, School of Medicine and Health, TUM, Munich, Germany; [5]Institute for Biomagnetism and Biosignalanalysis, University of Münster, Münster, Germany

## eLife Assessment

Gil Ávila et al. evaluated the aperiodic component in the medial prefrontal cortex using resting-state EEG recordings from 149 individuals with chronic pain and 115 healthy participants. The authors present **compelling** evidence that the aperiodic component of the EEG does not differentiate between those with chronic pain and healthy individuals. The study was well-designed and rigorously conducted, and the clear and conclusive results provide **important** insights that can guide future research in the field of pain neuroscience.

*For correspondence:
markus.ploner@tum.de

**Abstract** Chronic pain is a prevalent and debilitating condition whose neural mechanisms are incompletely understood. An imbalance of cerebral excitation and inhibition (E/I), particularly in the medial prefrontal cortex (mPFC), is believed to represent a crucial mechanism in the development and maintenance of chronic pain. Thus, identifying a non-invasive, scalable marker of E/I could provide valuable insights into the neural mechanisms of chronic pain and aid in developing clinically useful biomarkers. Recently, the aperiodic component of the electroencephalography (EEG) power spectrum has been proposed to represent a non-invasive proxy for E/I. We, therefore, assessed the aperiodic component in the mPFC of resting-state EEG recordings in 149 people with chronic pain and 115 healthy participants. We found robust evidence against differences in the aperiodic component in the mPFC between people with chronic pain and healthy participants, and no correlation between the aperiodic component and pain intensity. These findings were consistent across different subtypes of chronic pain and were similarly found in a whole-brain analysis. Their robustness was supported by preregistration and multiverse analyses across many different methodological choices. Together, our results suggest that the EEG aperiodic component does not differentiate between people with chronic pain and healthy individuals. These findings and the rigorous methodological approach can guide future studies investigating non-invasive, scalable markers of cerebral dysfunction in people with chronic pain and beyond.

## Introduction

Chronic pain is a highly prevalent disease (*Kennedy et al., 2014*) that severely decreases the quality of life of those who live with it, and imposes a substantial burden on society and healthcare systems (*Rice et al., 2016*). The neural mechanisms of chronic pain are incompletely understood, and effective therapies are lacking (*Cohen et al., 2021*). Therefore, advancing our understanding, assessment, and treatment of chronic pain is urgently needed.

Human and animal studies have revealed structural and functional changes in an extended network of brain areas in chronic pain, including the medial prefrontal cortex (mPFC; *Baliki and Apkarian, 2015*; *Kuner and Flor, 2017*). Cellular and microcircuit studies in rodent models of chronic pain have indicated imbalances between neural excitation and inhibition in this network (*Bliss et al., 2016*; *Kummer et al., 2020*; *Shiers and Price, 2020*; *Tan and Kuner, 2021*). Correspondingly, positron emission tomography (PET) and magnetic resonance spectroscopy (MRS) studies in humans have suggested alterations in the excitation/inhibition (E/I) balance in experimental and chronic pain (*Pasanta et al., 2023*; *Peek et al., 2020*; *Zhao et al., 2017*). Furthermore, changes in E/I likely relate to central sensitization (*Woolf, 2011*), which has been observed in many chronic pain states, including nociplastic pain conditions (*Kaplan et al., 2024*). Thus, assessing cerebral excitation and inhibition promises insights into the neural mechanisms of chronic pain. Moreover, such insights might aid in

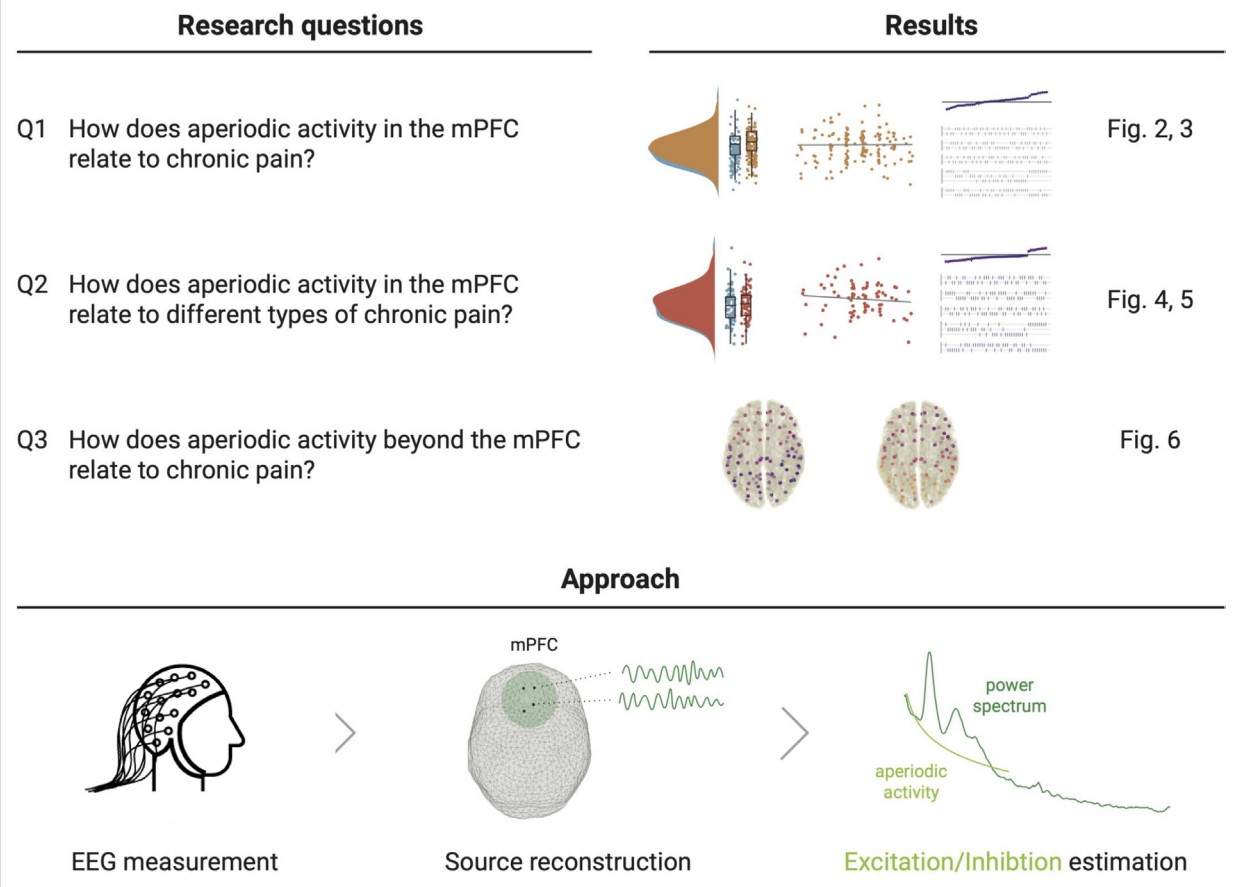

**Figure 1.** Study outline. The upper panel states the research questions and sketches their accompanying results from the two pre-registered analyses and their complementing multiverse analyses. The lower panel illustrates the pre-registered approach to estimate E/I non-invasively with EEG. We reconstructed the time series at one hundred spatial points in source space and summarized the medial prefrontal cortex (mPFC) activity (**Q1 and Q2**) by averaging power spectra in this region. We then estimated the aperiodic activity of the average power spectrum. The aperiodic exponent (the slope of the aperiodic activity in log-log space) is the proposed proxy for E/I, with high aperiodic exponents indicating low E/I ratio. *Figure 1—figure supplement 1* shows the grand averaged power spectra in the mPFC.

The online version of this article includes the following figure supplement(s) for figure 1:

**Figure supplement 1.** Grand averaged power spectra in the mPFC.

developing clinically useful biomarkers of chronic pain. However, non-invasive, scalable, whole-brain measurements of the E/I balance in people with chronic pain have not been performed so far.

Recently, novel electroencephalography (EEG) measures have been proposed as non-invasive proxies for E/I with high translational potential (*Ahmad et al., 2022*). Specifically, the aperiodic exponent of the EEG power spectrum has been highlighted as a measure of the E/I ratio (*Gao et al., 2017*; *Lombardi et al., 2017*). The power spectrum of EEG signals (*Figure 1*) typically features narrow band peaks over a power distribution following a 1/f$^\chi$ shape, with power decreasing linearly as frequency increases on a log-log scale (*Buzsáki and Draguhn, 2004*; *Miller et al., 2009*). Traditional EEG analyses have focused on oscillatory activity (peaks in the power spectrum) and activity at specific frequency bands. However, recent studies have highlighted the physiological significance of the power spectrum's aperiodic (1/f$^\chi$) component (*Donoghue et al., 2020*; *Lendner et al., 2020*; *Ostlund et al., 2021*; *Tröndle et al., 2022*). In particular, the slope of the aperiodic component, given by the aperiodic exponent ($\chi$), has been linked to the E/I balance. Higher exponents (steeper spectra) indicate a shift toward inhibition, and lower exponents toward excitation (*Gao et al., 2017*; *Lendner et al., 2020*). Moreover, E/I disruptions have been proposed as a common feature underlying different neuropsychiatric disorders (*Ahmad et al., 2022*; *Sohal and Rubenstein, 2019*). Accordingly, first EEG studies assessing the aperiodic component in several brain disorders have revealed changes in schizophrenia (*Molina et al., 2020*), ADHD (*Ostlund et al., 2021*; *Robertson et al., 2019*), and Alzheimer's disease (*Martínez-Cañada et al., 2023*).

In the present study, we characterized the E/I balance in people with chronic pain non-invasively. To this end, we analyzed the aperiodic component of resting-state EEG recordings in a large cohort of people with chronic pain (N=149), compared it to healthy participants (N=115), and related it to pain intensity. We hypothesized that changes in the aperiodic exponent would particularly occur in the medial prefrontal cortex (mPFC), as this is a central hub for pain processing and E/I imbalances have been observed in this area (*Bliss et al., 2016*; *Kummer et al., 2020*; *Shiers and Price, 2020*; *Tan and Kuner, 2021*). We complemented this region-of-interest analysis with a whole-brain analysis of the aperiodic component. Furthermore, we investigated aperiodic exponents in subtypes of chronic pain.

The current study rigorously pursued open and reproducible science practices. We pre-registered the study (https://osf.io/xshmy), performed blind analyses with Bayesian hypothesis testing, and made all data and code openly available (*Hardwicke and Wagenmakers, 2023*; *MacCoun and Perlmutter, 2015*; *Wagenmakers et al., 2018*). Additionally, we conducted a multiverse analysis to assess the robustness of the results across different analytical choices (*Simonsohn et al., 2020*; *Steegen et al., 2016*). This is particularly relevant as aperiodic activity is an emerging metric for which the influence of several methodological decisions is not fully clear.

Together, the study is intended to represent a step forward in developing translational, non-invasive, and robust tools to better understand chronic pain mechanisms. Such insights might pave the way for developing clinically useful biomarkers and novel pharmacological and neuromodulatory treatments for chronic pain.

## Results

To investigate the E/I balance in chronic pain, we analyzed the aperiodic exponent of the power spectrum in 149 people with different types of chronic pain and 115 healthy participants. The aperiodic component was characterized in eyes-closed resting-state EEG recordings following a pre-registered approach (https://osf.io/xshmy). Specifically, we addressed three research questions (*Figure 1*). We first studied how the aperiodic activity in the mPFC relates to a mixed sample of people with chronic pain (Q1). We further asked how aperiodic activity in the mPFC relates to different types of chronic pain (Q2). Finally, we assessed the relationship between aperiodic activity and chronic pain beyond the mPFC in a whole-brain analysis (Q3). For each question, we performed two pre-registered analyses. First, we analyzed whether aperiodic exponents differ between people with chronic pain and healthy participants. Second, we analyzed whether aperiodic exponents in people with chronic pain correlate with their average pain intensity.

We investigated the research questions in a two-step approach. We first estimated the aperiodic activity and conducted the two analyses according to the pre-registration. As a second step, we conducted a non-pre-registered multiverse analyses to test the robustness of the results, since the aperiodic component is an emerging metric, and it is unclear how different methodological decisions

**Table 1.** Analytical decisions and parameters investigated in the multiverse analysis.

Parameters in italics are the ones used in the pre-registered analysis. In total 48 different combinations of parameters (specifications) were investigated.

| Analytical decision | Parameter | Description |
|---|---|---|
| Epoch length | *2 s* | Length of the epochs in seconds. Epoch length affects the frequency resolution of the power spectrum. '2 s' implies a frequency resolution of 0.5 Hz, '5 s' of 0.2 Hz. |
| | 5 s | |
| Taper | *dpss* | Method to estimate the power spectrum. 'Dpss' performs frequency analysis with multiple dpss tapers and 1 Hz spectral smoothing. 'Hanning' performs frequency analysis with a single hanning taper. |
| | hanning | |
| Avg. psd | *yes* | Method to summarize aperiodic activity on the mPFC. 'Yes' averages the power spectra of all the source locations encompassing the mPFC and derives the aperiodic exponent from this average. 'No' models the power spectrum for each location and averages aperiodic parameters obtained from each power spectra. |
| | no | |
| Fooof range | *2–40 Hz* | Frequency range in which to model the power spectrum. '2–40 Hz' is the pre-set value of the 'spectparam algorithm' (*Donoghue et al., 2020*). '40–60' is the original setting of *Gao et al., 2017*. '1–100' includes the whole power spectrum. |
| | 40–60 Hz | |
| | 1–100 Hz | |
| Fooof knee | *no* | Parameter that controls the bend of the model of aperiodic activity. 'No' omits the estimation of the knee. 'Yes' models the knee. |
| | yes | |

affect its estimation. Specifically, we performed specification curve analyses (*Simonsohn et al., 2020*), a type of multiverse analysis that includes the identification of theoretically and statistically valid ways of analyzing the data (specifications), the graphical display of evidence in a descriptive specification curve, and inference tests across specifications to reach an overall conclusion on the results. One specification thereby refers to one combination of analytical settings, i.e., one version of the analysis.

We identified five methodological decisions that could influence the estimation of aperiodic activity, each with two or three equally valid parameter settings (*Table 1*, Methods). The combination of all these parameter settings yielded 48 different specifications, including the specification of the pre-registered analyses. Thus, in each multiverse analysis, we estimated the aperiodic exponent in 48 different ways and repeated the corresponding pre-registered analysis. The result is a specification curve, depicting in the x-axis the specifications, sorted by increased effect size, in the y-axis the effect size, and color-coded the degree of evidence ($BF_{10}$). We also quantified how likely the observed specification curve was, compared to a null distribution of curves indicating no effect with three inference tests (*Simonsohn et al., 2020*). We performed multiverse analyses for research questions Q1 and Q2. We did not perform multiverse analysis for Q3, as it would yield a higher number of results than what can be statistically addressed and presented (please refer to Methods).

## How does aperiodic activity in the mPFC relate to chronic pain?

We first compared the aperiodic exponents in the mPFC between the entire sample of 149 people with chronic pain and 115 healthy participants. We controlled the potential confound of age in aperiodic exponents by regressing age out from all the studied variables. Average aperiodic exponents before age correction were 1.10±0.16 (mean ± std) for the healthy participants and 1.10±0.18 for the participants with chronic pain. The results provided moderate evidence against a difference between groups (*Figure 2A*, Cohen's d=−0.011, $BF_{10}$=0.136, Bayesian two-sided independent samples t-test). The specification curve analysis confirmed this finding (*Figure 2B*). All specifications indicated moderate evidence against a difference in aperiodic exponents between groups (all $BF_{10}$ <0.22). Statistical inference on the specification curve indicated that the curve was not significantly different from a null distribution of specification curves ($p_{median}$ = 0.89, $p_{share}$ = 1, $p_{aggregate}$ = 0.95, please refer to the Methods for a description of the test statistics). Thus, both pre-defined and multiverse analysis provided robust evidence against a difference in aperiodic exponents in the mPFC between healthy participants and people with chronic pain.

Next, we correlated the aperiodic exponents in the mPFC of people with chronic pain with their average pain intensity. The results provided moderate evidence against a correlation (*Figure 3A*, Pearson's R=0.006, $BF_{10}$=0.103, Bayesian correlation test). The specification curve analysis confirmed

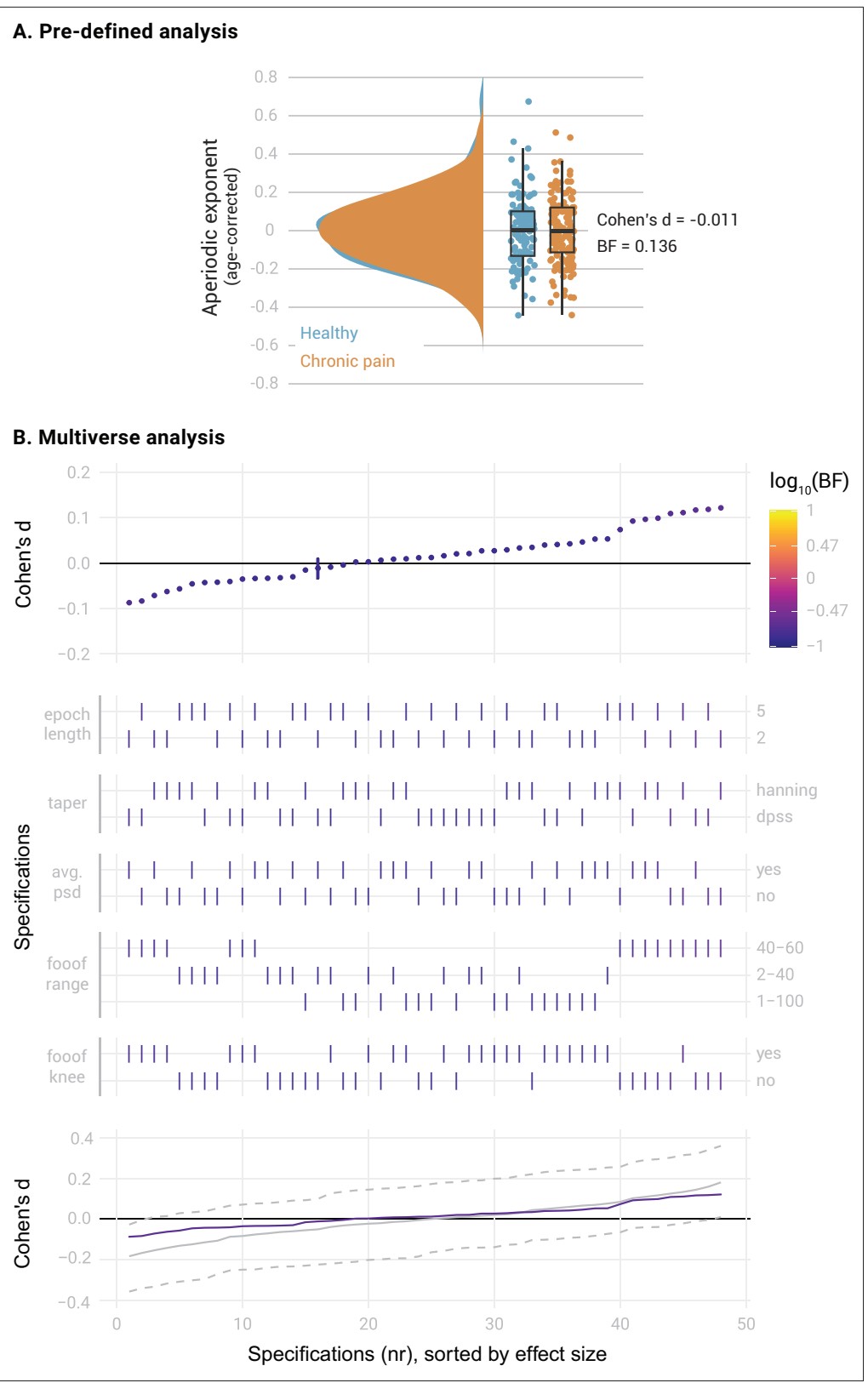

**Figure 2.** Comparison of the aperiodic exponent in the medial prefrontal cortex (mPFC) between people with chronic pain (N=149) and healthy participants (N=115). (**A**) Pre-defined analysis. Raincloud plots include individual data points, boxplots and probability density functions. The box of the boxplots ranges from the first to the third quartile, with a thicker line representing the sample median. Whiskers extend the first and third quartile to 1.5

*Figure 2 continued*

times the interquartile range. Age was regressed out from the aperiodic exponents of each group. Bayes Factor (BF) was derived from a two-sided independent samples Bayesian t-test comparing the two groups. (**B**) Multiverse analysis. The upper panel visualizes the specification curve, which indicates the effect size (y axis) and evidence (color coded) for each specification (x axis). Specifications are ordered by effect size. A vertical bar indicates the specification of the pre-defined analysis. The color scale's upper and lower limits indicate strong evidence for and against a difference between groups. The middle panel further describes the parameters of each specification, with vertical bars indicating the parameters selected in each methodological decision (*Table 1*). The lower panel visualizes inference on the specification curve. In purple, the original specification curve is depicted. In grey, the null distribution of specification curves derived from 500 randomized curves is represented by the median curve (solid grey line) and 2.5th and 97.5th percentile curves (dashed lines). Intuitively, if the original curve lies between the 2.5th and 97.5th percentile curves, it is not significantly different from the null distribution of specification curves.

this result (*Figure 3B*). All specifications indicated moderate evidence against a correlation between pain ratings and aperiodic exponents (all $BF_{10}$ <0.26). Statistical inference on the specification curve indicated that the curve was not significantly different from a null distribution of specification curves ($p_{median}$ = 0.63, $p_{share}$ = 1, $p_{aggregate}$ = 0.74). Apart from the aperiodic exponent, the aperiodic offset is another parameter estimated when computing the aperiodic component. The aperiodic offset has a less clear physiological interpretation than the aperiodic exponent. Nevertheless, we also investigated whether it was altered in the mPFC in people with chronic pain or was related to pain intensity (*Figure 3—figure supplement 1*). We found moderate evidence against a difference in aperiodic offsets between people with chronic pain and healthy participants (Cohen's d=–0.03, $BF_{10}$=0.14). Furthermore, we found moderate evidence against a correlation between aperiodic offsets and average pain ratings (R=0.06, $BF_{10}$=0.14). Moreover, we exploratively investigated the effects of age on aperiodic exponents in the mPFC in our dataset. The results are shown in *Figure 3—figure supplement 2*.

In summary, we found evidence against a difference in aperiodic activity in the mPFC between healthy participants and a diverse cohort of people with chronic pain. We also found evidence against a correlation between aperiodic exponents and pain intensity in people with chronic pain. Multiverse analyses confirmed the robustness of these findings across different analytical parameters.

## How does aperiodic activity in the mPFC relate to different types of chronic pain?

The sample of people with chronic pain included participants with different types of chronic pain. Thus, effects specific to certain subtypes of pain might have been missed. Therefore, we repeated pre-registered and multiverse analyses for the two largest clinical subgroups, people with Chronic Back Pain (CBP, N=80) and people with Chronic Widespread Pain (CWP, N=33). We randomly drew a subsample of healthy participants of equal size for each subgroup, matched in age, gender, and dataset to which they belonged (see Methods). Average aperiodic exponents before age correction were 1.11±0.19 for the people with CBP and 1.02±0.13 for the people with CWP.

In people with CBP (*Figure 4*), we found moderate evidence against a difference between the mPFC aperiodic exponents of people with CBP and healthy participants (Cohen's d=–0.12, $BF_{10}$=0.23). The specification curve analysis confirmed this result across 46 specifications, with only two showing inconclusive evidence (0.17 < $BF_{10}$<0.36). Statistical inference on the specification curve indicated that the curve was not significantly different from a null distribution of specification curves ($p_{median}$ = 0.59, $p_{share}$ = 1, $p_{aggregate}$ = 0.61). There was also moderate evidence against a correlation between the aperiodic exponent and pain intensity in people with CBP (Pearson's R=–0.09, $BF_{10}$=0.19). The specification curve analysis indicated that this result was robust, with 44 specifications showing evidence in favor of the null hypothesis and only four showing inconclusive evidence (0.14 < $BF_{10}$<0.42). Statistical inference on the specification curve indicated that the curve was not significantly different from a null distribution of specification curves ($p_{median}$ = 0.23, $p_{share}$ = 1, $p_{aggregate}$ = 0.35).

In people with CWP (*Figure 5*), we found inconclusive evidence regarding a difference in the mPFC aperiodic exponent between people with CWP and healthy participants (Cohen's d=0.26, $BF_{10}$=0.41). The specification curve analysis revealed that approximately 1/3 of the specifications supported

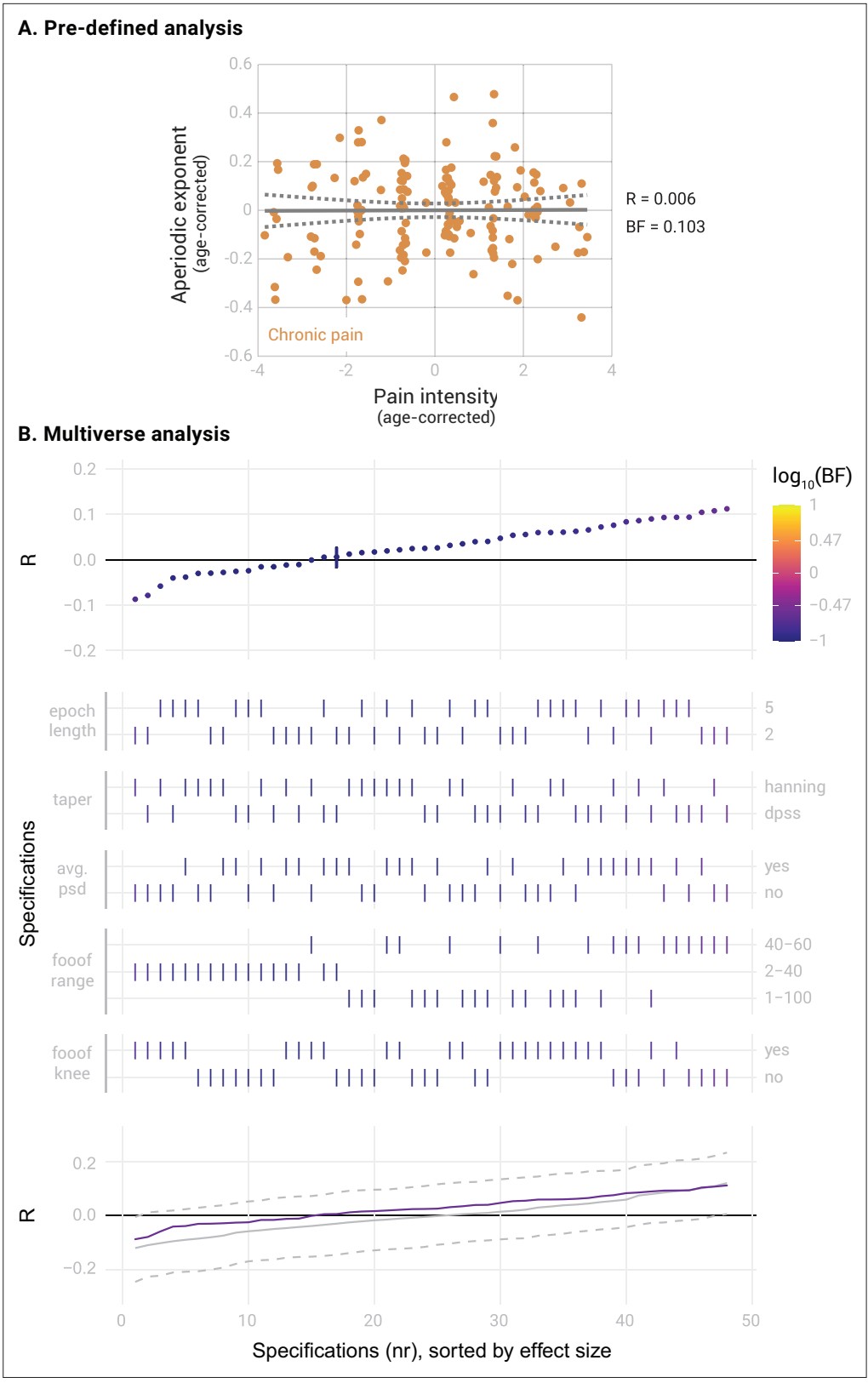

**Figure 3.** Correlation between the aperiodic exponent in the medial prefrontal cortex (mPFC) and average pain intensity in people with chronic pain (N=149). (**A**) Pre-defined analysis. The scatter plot represents the aperiodic exponent and average pain intensity ratings. Age was regressed out from both variables to perform a partial correlation. Grey lines indicate the linear regression slope (solid line) and the 95% confidence interval (dashed

*Figure 3 continued*

lines). Bayes Factor (BF) was obtained from a Bayesian linear correlation performed on the residuals. (**B**) Multiverse analysis including the specification curve panel with Pearson's correlation coefficients (**R**) as effect size, the specifications panel, and the inference panel as in *Figure 2*.

The online version of this article includes the following figure supplement(s) for figure 3:

**Figure supplement 1.** Relationship between the aperiodic offset in the mPFC and chronic pain.

**Figure supplement 2.** Relationship between the aperiodic exponent in the mPFC and age for all participants in the study (N=264).

---

the null hypothesis (specifications 7–26, $0.25 < BF_{10} < 0.33$), while 2/3 of the specifications showed inconclusive evidence (specifications 1–6 and 27–48, $0.34 < BF_{10} < 0.83$). Statistical inference on the specification curve indicated that the curve was not significantly different from a null distribution of specification curves ($p_{median} = 0.31$, $p_{share} = 1$, $p_{aggregate} = 0.41$). Furthermore, we found moderate evidence against a correlation between the aperiodic exponent and pain intensity in people with CWP ($R=0.03$, $BF_{10}=0.22$). The specification curve analysis provided further evidence against a correlation in approximately half of the specifications (specifications 22–48, $0.22 < BF_{10} < 0.33$). However, 20 specifications showed inconclusive evidence (specifications 2–21, $0.33 < BF_{10} < 1.1$), and one specification indicated a negative correlation between aperiodic exponents and pain (specification 1, $BF_{10}=3.1$). Statistical inference on the specification curve indicated that the curve was not significantly different from a null distribution of specification curves ($p_{median} = 0.25$, $p_{share} = 0.17$, $p_{aggregate} = 0.49$).

In summary, we found moderate evidence against altered aperiodic exponents in the mPFC and against a correlation between aperiodic exponents and pain intensity in people with chronic back pain. However, we found inconclusive evidence for the role of aperiodic exponents in the mPFC in people with chronic widespread pain.

## How does aperiodic activity beyond the mPFC relate to chronic pain?

So far, we focused on the medial prefrontal cortex due to previous evidence for its relevance to chronic pain (*Bliss et al., 2016*; *Kummer et al., 2020*; *Shiers and Price, 2020*; *Tan and Kuner, 2021*). However, pain is a network phenomenon involving multiple brain areas. We, therefore, extended our analyses beyond the mPFC and investigated aperiodic exponents at a whole-brain level (*Figure 6*).

We estimated aperiodic exponents using the pre-defined settings from the previous analyses in 100 points of the MNI template. Brain locations were the centroid regions of the 100-parcel version of the Schaefer atlas with 17 networks (*Schaefer et al., 2018*; *Yeo et al., 2011*). To assess differences in the aperiodic exponent between healthy participants and people with chronic pain, we performed independent sample two-sided t-tests between groups at every region. As there are no standards for correcting Bayesian statistics for multiple comparisons, we performed frequentist t-tests and adjusted the p-values with the resampling-based FDR correction (*Yekutieli and Benjamini, 1999*). We did not find a difference in aperiodic exponents between healthy participants and people with chronic pain in any of the 100 regions (all $p_{adj} > 0.05$, $-0.39 < $ Cohen's $d < 0.06$). Raw values of the age-corrected aperiodic exponents of people with chronic pain and healthy participants are available in *Figure 7*.

To investigate the relationship between the aperiodic exponent and pain intensity in people with chronic pain beyond the mPFC, we performed a Pearson's correlation between the two variables at each region of interest. Again, we did not observe any significant correlation after multiple comparisons correction with resampling-based FDR (all $p_{adj} > 0.05$, $-0.11 < R < 0.22$). Please note that no whole-brain multiverse analyses were conducted as the group comparisons and correlations would include 100 multiverse analyses each.

In summary, we did not observe any difference in aperiodic exponents between healthy participants and people with chronic pain at a whole brain level. We also did not observe a correlation between aperiodic exponents and pain intensity in any brain region beyond the mPFC.

## Discussion

In the present study, we non-invasively characterized the balance between cerebral excitation and inhibition in people with chronic pain. To this end, we analyzed the aperiodic component of resting-state EEG data in a large cohort of people with chronic pain, compared it to healthy participants, and

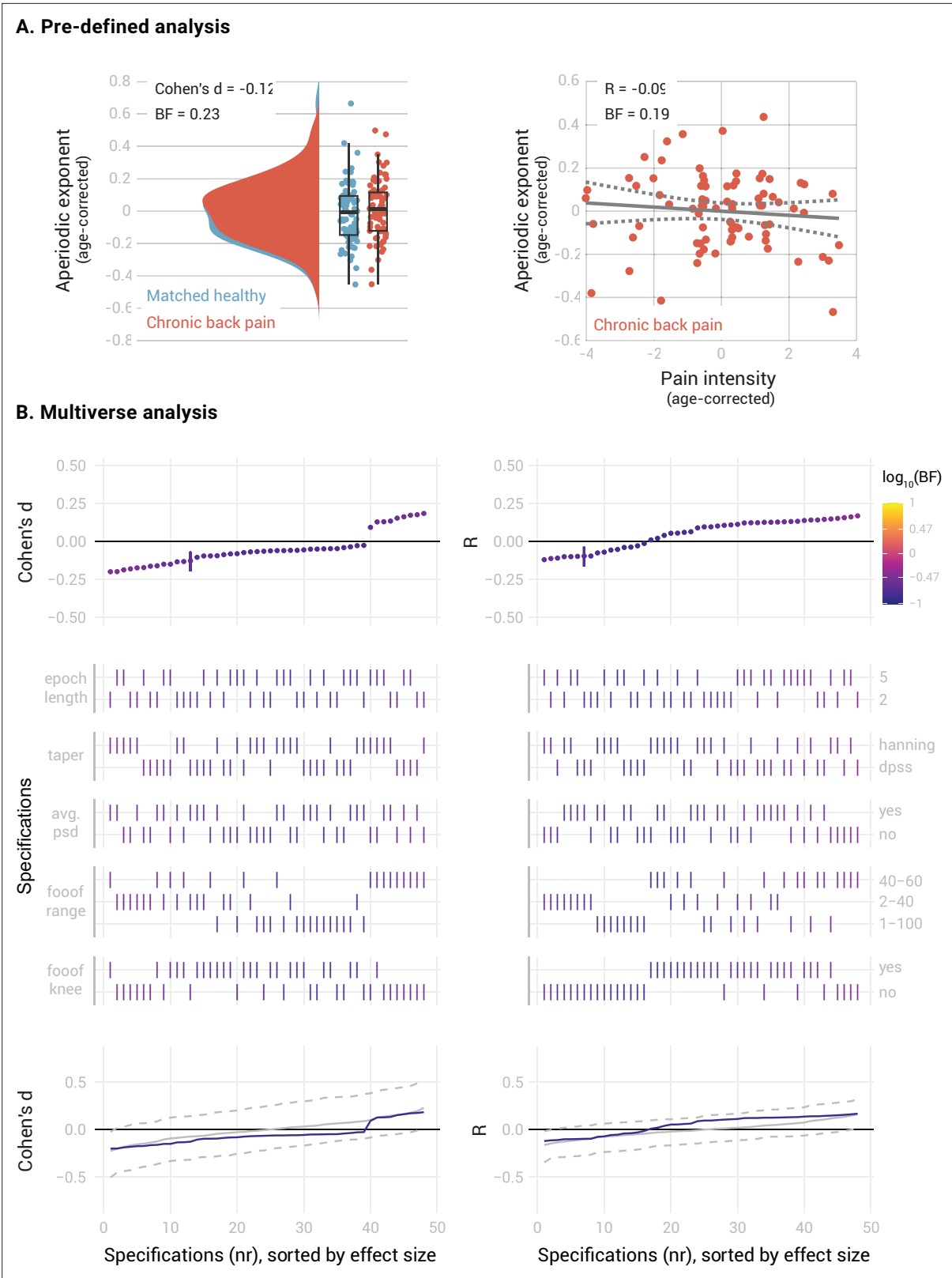

**Figure 4.** elationship between the aperiodic exponent in the mPFC and average pain intensity in people with Chronic Back Pain. (**A**) Pre-defined analysis. (**B**) Multiverse analysis. The left column shows differences between people with chronic back pain (N=80) and a healthy subsample, matched in sample size, age, gender, and dataset belonging to the CBP sample (N=80). The right column shows the correlation between aperiodic exponents and pain intensity in people with CBP (N=80). Aperiodic exponents were corrected for age by regressing out age from the participants subsample

*Figure 4 continued on next page*

*Figure 4 continued*

(left panel) or the people with CBP sample (right panel). Pain intensity ratings were corrected for age by regressing out age from the people with CBP subsample.

related it to pain intensity. Our results did not show differences in the aperiodic component between people with chronic pain and healthy participants nor relationships between the aperiodic component and pain intensity. These findings are strengthened by preregistration and Bayesian statistics, which allow the quantification and interpretation of both positive and negative results. Multiverse analyses confirmed the robustness of the results across a wide range of analytical decisions. In this way, the findings enhance the understanding of the brain mechanisms of chronic pain and can guide future research on developing non-invasive markers of cerebral excitability in chronic pain and other brain disorders.

The observed lack of a relationship between the aperiodic component and chronic pain can be due to several reasons. First, there is indeed no relationship between cerebral E/I and chronic pain. Considering the extensive evidence for pain-related changes in cerebral E/I in animals and humans, this explanation appears unlikely. In animal models of chronic pain, numerous studies have found alterations of E/I, mainly in the prefrontal cortex, but also in other cortical and subcortical brain areas (*Bliss et al., 2016*; *Kummer et al., 2020*; *Shiers and Price, 2020*; *Tan and Kuner, 2021*). In humans, magnetic resonance spectroscopy findings show abnormal glutamate and GABA signaling in cortical and subcortical regions in people with chronic pain (*Peek et al., 2020*; *Zhao et al., 2017*).

Second, there is a relationship between E/I and chronic pain, but it cannot be detected using EEG. This explanation appears reasonable since excitability changes in chronic pain might differ across cell types and brain regions (*Bliss et al., 2016*; *Kummer et al., 2020*; *Shiers and Price, 2020*; *Tan and Kuner, 2021*). Thus, possible E/I effects in chronic pain might be too subtle or too heterogeneous to be detectable at the scalp using EEG. However, EEG findings in brain disorders known to be associated with E/I alterations argue against this possibility (*Martínez-Cañada et al., 2023*; *Molina et al., 2020*; *Ostlund et al., 2021*; *Robertson et al., 2019*). Moreover, we have analyzed the aperiodic component exclusively in source space. However, source space analyses can yield imprecise results, particularly when default volume conduction models and electrode coordinates are used. A systematic exploration of different source space approaches and analyses in electrode space might address this issue.

Third, there is a relationship between E/I and chronic pain detectable with EEG, but aperiodic activity is an inappropriate measure of E/I. Although this explanation cannot be ruled out, a relationship between aperiodic activity and E/I has been supported by in-vivo electrophysiological recordings in rodents (*Gao et al., 2017*), computational models (*Gao et al., 2017*; *Lombardi et al., 2017*; *Martínez-Cañada et al., 2023*), the gabaergic drug propofol (*Lendner et al., 2020*), and neuromodulation studies (*Martínez-Cañada et al., 2023*). Cross-modal integration of findings from PET, MRS, and EEG studies would be desirable to strengthen the link between aperiodic EEG activity and E/I. Furthermore, future studies might also assess non-linear relationships between chronic pain and EEG measures of E/I. In addition, other EEG measures of E/I have been proposed (*Ahmad et al., 2022*) and could be explored in future studies.

Fourth, there is a relationship between E/I measured with the EEG aperiodic exponent and chronic pain, but our heterogeneous sample of people with chronic pain has hampered the detection of a small effect. It is plausible that E/I imbalances are more prominent in specific subtypes of chronic pain. For example, clinical observations suggest that changes in excitability are particularly relevant in chronic widespread pain (CWP) (*Sluka and Clauw, 2016*) and migraine (*O'Hare et al., 2023*), but possibly less in other chronic pain conditions. Our findings are compatible with this possibility, as they showed robust negative evidence in people with CBP, but partially inconclusive evidence in our small sample of people with CWP. Thus, future studies might include larger sample sizes and focus on specific subtypes of chronic pain, for example CWP or migraine. They might also include a standardized and more extensive assessment of clinical measures, as well as the analysis of EEG recordings with eyes open. Moreover, such studies should be designed and sufficiently powered to assess the effects of gender and medication on the aperiodic EEG component.

Considering the negative findings of the present study, does it make sense to further investigate the role of the aperiodic component of resting-state EEG in chronic pain? The possible gain and

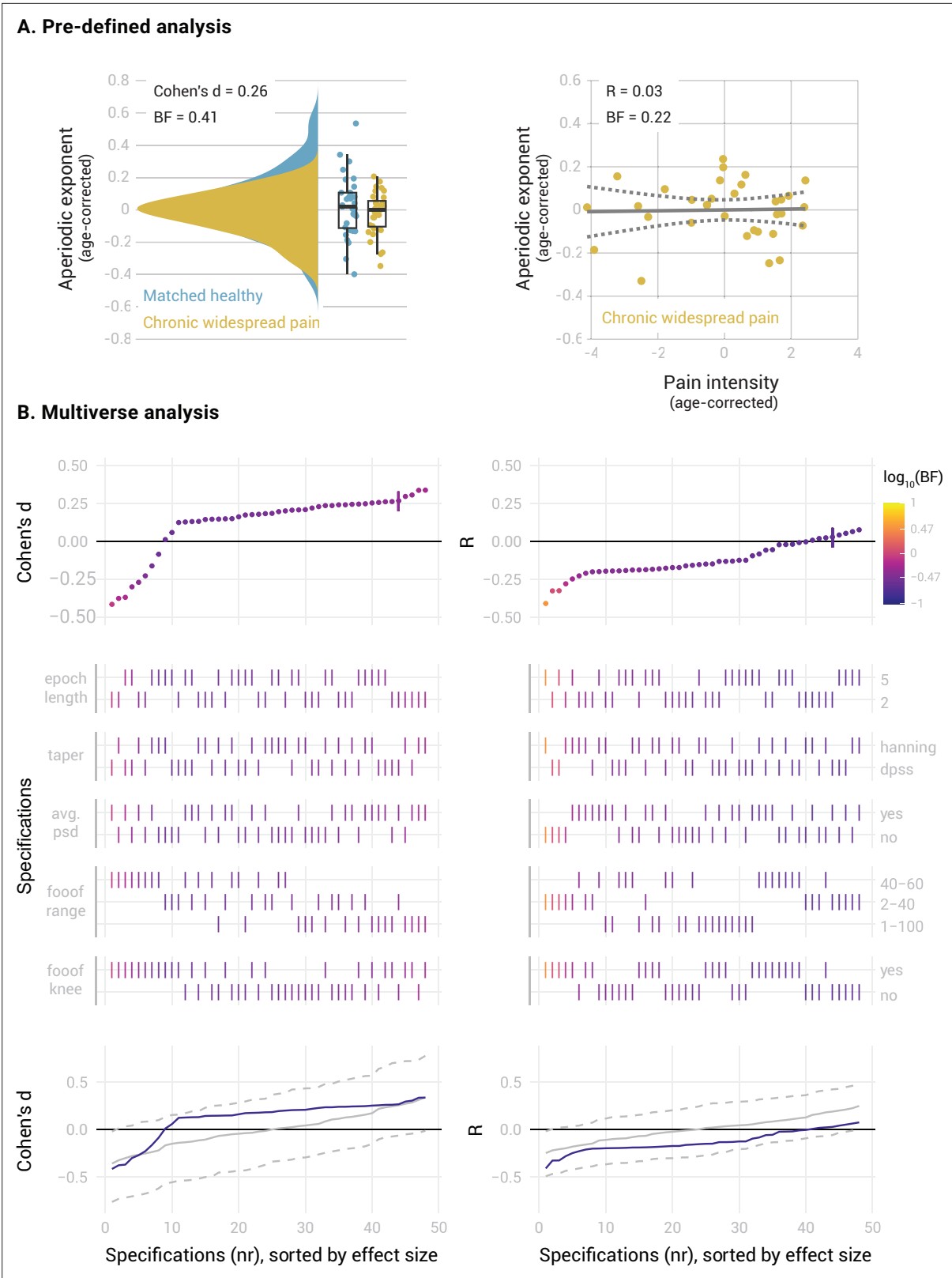

**Figure 5.** Relationship between the aperiodic exponent in the mPFC and average pain intensity in people with Chronic Widespread Pain. (**A**) Pre-defined analysis. (**B**) Multiverse analysis. The left column shows differences between people with Chronic Widespread Pain (N=33) and a healthy subsample matched in sample size, age, gender, and dataset belonging to the CWP sample (N=33). The right column shows the correlation between aperiodic exponents and pain intensity in people with CWP (N=33). Aperiodic exponents were corrected for age by regressing out age from the

*Figure 5 continued on next page*

*Figure 5 continued*

participants subsample (left panel) or the people with CWP sample (right panel). Pain intensity ratings were corrected for age by regressing out age from the people with CWP subsample.

the necessary efforts might guide this decision. We consider the potential gain high. Finding a non-invasive measure of cortical excitability, a critical brain mechanism of chronic pain related to central sensitization (*Woolf, 2011*) and nociplastic pain (*Kaplan et al., 2024*), would significantly advance our understanding of the disease. Moreover, it could help to develop a broadly available, cost-efficient, and scalable biomarker. Such a biomarker could fulfill clinically valuable functions such as determining the risk/susceptibility to develop chronic pain or the prognosis of the disease and predicting treatment responses (*Davis et al., 2017*). A biomarker close to an essential neural mechanism of chronic pain, that is increased excitability, loss of inhibition, and central sensitization, is particularly promising. These arguments must be weighed against the efforts needed to obtain further insights into the role of the aperiodic EEG in chronic pain. EEG is broadly available, and the efforts to perform EEG recordings are decreasing with progress in EEG hardware with dry electrodes (*Ng et al., 2022*) and mobile use (*Niso et al., 2023*). Thus, even when aiming for large sample sizes, as opposed to other non-invasive measures of brain activity, the efforts are comparably low. Taken together the high possible gain and the comparatively low efforts, it might be worth further pursuing this line of research.

The present findings might also have implications for research beyond chronic pain. The coordination of excitatory and inhibitory brain activity is a fundamental property of brain function (*Turrigiano and Nelson, 2004*), and the E/I balance has been shown to play an important role in various brain functions and disorders. For instance, E/I imbalances have been frequently observed in autism (*Sohal and Rubenstein, 2019*) and schizophrenia (*Liu et al., 2021*). Research on the E/I balance in all these domains could benefit from methodological exchange, integration, and harmonization across fields and disorders. The present approach might represent an example of how a human EEG study can be designed to make positive as well as negative findings most valid and informative.

In conclusion, our study did not find a relationship between the aperiodic component of resting-state EEG and chronic pain, supported by high standards of open and reproducible science. These findings should guide, rather than discourage, further research on possible EEG biomarkers of chronic pain. Future studies might focus on specific chronic pain subtypes and aim at larger, multi-site samples. By pursuing this approach, we may identify a non-invasive and scalable marker of altered cerebral excitability in chronic pain. Eventually, this might help to develop biomarkers and even new

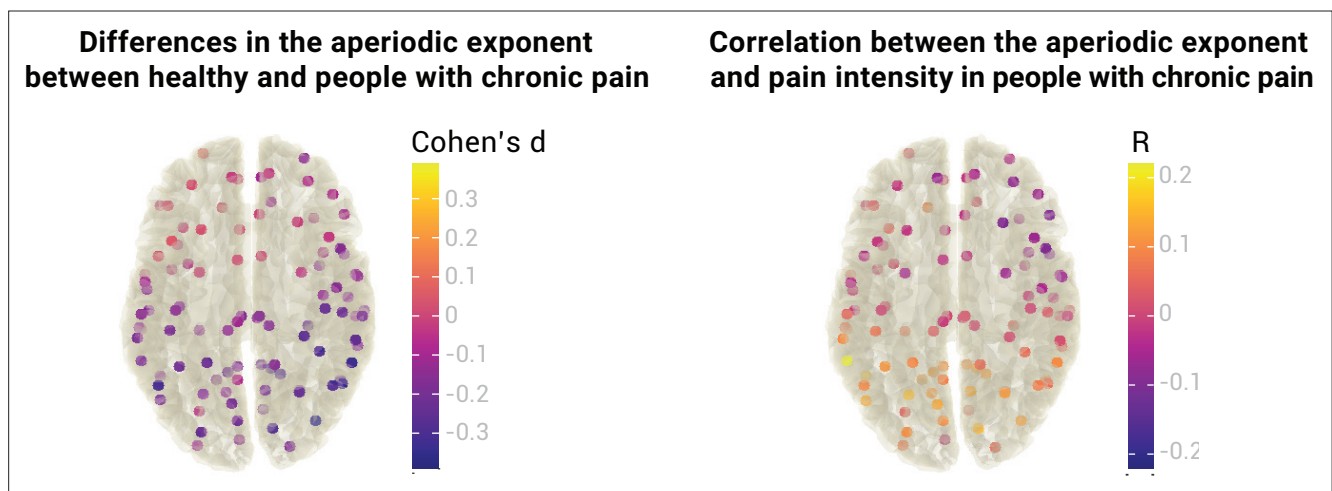

**Figure 6.** Relationship between the aperiodic exponent and chronic pain at a whole-brain level. The left panel visualizes differences in the aperiodic exponent (Cohen's d) between healthy participants (N=115) and people with chronic pain (N=149) at 100 different brain locations. The right panel visualizes the correlation between the aperiodic exponent and average pain intensity (Pearson's correlation coefficient, R) in people with chronic pain at 100 different brain locations. No statistical differences or correlations were found in any location after correcting for multiple comparisons. Aperiodic exponents and pain ratings were corrected for age.

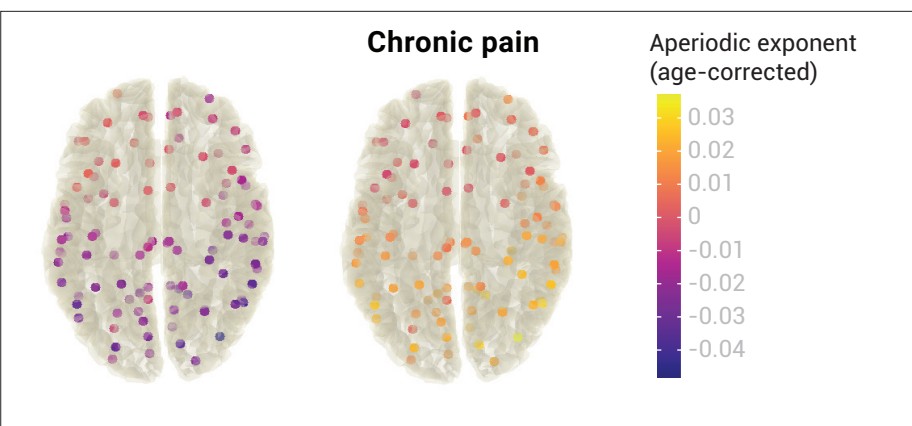

**Figure 7.** Aperiodic exponents, corrected for age, of healthy participants and people with chronic pain across the whole brain.

therapeutic targets and endpoints aiming at restoring the E/I balance (*Martínez-Cañada et al., 2023*; *van Bueren et al., 2023*).

## Methods
### Study design
We investigated the relationship between E/I and pain using three public datasets acquired at the PainLabMunich, Technical University of Munich, Germany. The datasets included resting-state EEG data and clinical and demographic variables of 151 people with chronic pain and 115 healthy participants. Two datasets were already published and were re-analyzed. The third dataset was not analyzed so far and has been made publicly available for this project (https://osf.io/qgfma/). As each dataset was initially acquired for a different project, the EEG recording paradigm and the collected clinical and demographic variables slightly differed between datasets. We refer the reader to the original publications for further details regarding inclusion/exclusion criteria of participants, administered questionnaires, and technical characteristics of the EEG recordings (*Heitmann et al., 2022*; *May et al., 2019*; *Ta Dinh et al., 2019*; *Tiemann et al., 2012*).

This study was pre-registered in OSF Registries on 11 May 2023 (https://osf.io/xshmy) as a secondary data pre-registration. In the pre-registration, we hypothesized (1) differences in aperiodic activity in the mPFC between people with chronic pain and healthy participants and (2) a relationship between aperiodic activity in the mPFC and pain in people with chronic pain. We performed these two pre-registered analyses for our three research questions (*Figure 1*). We complemented each pre-registered analysis with a multiverse analysis to check the robustness of the results across different methodological decisions.

### Participants
Dataset 1 contains cross-sectional data of 101 people with different chronic pain conditions and 88 healthy participants. The participants were recorded in three different cohorts. The first one included 20 people with chronic widespread pain and 22 healthy participants (*Tiemann et al., 2012*). The second one contained 34 people with chronic back pain (*May et al., 2019*). The third one contained 47 people with different types of chronic pain and 66 healthy participants (*Ta Dinh et al., 2019*). Dataset 2 is a longitudinal dataset comprising 50 participants with different types of chronic pain who underwent multimodal pain therapy (*Heitmann et al., 2022*). Only the first session ('baseline') is analyzed in the present study. Dataset 3 has not been previously analyzed or published and includes 27 healthy participants.

The gender ratio of both groups was similar (people with chronic pain: 98 females and 53 males, healthy: 74 females and 41 males). The total sample included participants of a wide age range (people with chronic pain: 86–18, healthy: 79–18) but mostly white men and women who were currently

pursuing or previously acquired higher education. Thus, the study sample constitutes a specific sample not necessarily representative of Germany or any region.

## Data acquisition

During data acquisition sessions, participants first completed clinical and demographic questionnaires. Then, brain activity was recorded using EEG during the resting state, i.e., participants were asked to stay relaxed and wakeful without doing any particular task.

EEG data was recorded with 64 passive electrodes in the standard 10/20 positions (Easycap, Hersching, Germany) and a BrainAmp MR plus amplifier (Brain Products GmbH, Gilching, Germany). For most participants, two blocks of 5 min each were recorded, one with eyes closed and one with eyes open. This study only analyzed the eyes closed condition, available for all participants. During recording, electrodes were referenced to FCz and grounded at AFz. Data were obtained at a sampling frequency of 1000 Hz and were band-pass filtered online between 0.016 and 250 Hz.

The following demographic and clinical variables were used in this study: (1) Age of the participant at the recording date in years. (2) Average pain intensity, defined as the average pain intensity during the past four weeks obtained with a numerical rating scale from the PainDetect questionnaire (0=no pain, 10=maximum pain). In the Chronic Widespread Pain subset of Dataset 1 (N=20), this information was not available, and the variable 'current pain at the time of the recording' was used instead (0=no pain, 10=maximum pain). (3) Clinical diagnosis.

## Data gathering, blinding, and preprocessing

The three datasets were merged into one, and participants' IDs were randomly reassigned in the form sub-XXX, where XXX indicated a number between 001 and 266. In this way, researchers were blinded to the initial dataset to which the participant belonged. One participant in Dataset 2 was excluded because they had already been included in Dataset 1. Another participant of Dataset 2 was excluded due to technical problems during the recording. Therefore, the final sample size was 149 people with chronic pain and 115 healthy participants.

Next, two versions of the demographic data were prepared. In one version, participants' group labels (chronic pain or healthy) were randomly re-assigned, keeping the ratio of the original labels. In the other version pain intensity scores were randomly re-assigned to people with chronic pain. All pre-registered analyses and statistical scripts were prepared using these modified versions of the demographic data. In this way, researchers were blinded during the computation and visualization of intermediate results, preventing p-hacking and confirmation bias (*Hardwicke and Wagenmakers, 2023*; *MacCoun and Perlmutter, 2015*). Only when the statistical and visualization scripts were fully developed were they executed with the original demographic data.

EEG data was automatically preprocessed in MATLAB (Mathworks, Natick, MA), with the default settings of the DISCOVER-EEG pipeline version 1.0.0. (*Gil Ávila et al., 2023*; *Pernet et al., 2020*) using the EEGLAB Toolbox (*Delorme and Makeig, 2004*). Briefly, raw data in the BIDS standard were loaded into EEGLAB and downsampled to 250 Hz. Line noise was removed with the CleanLine plug-in. Then, data were high-pass filtered at 0.5 Hz, and bad channels were rejected with the clean_rawdata function. Data was re-referenced to the average reference, and Independent Component Analysis with the 'runica' algorithm was performed. Independent components categorized as muscle or eye artifacts by the ICLabel classifier (*Pion-Tonachini et al., 2019*) were removed. Previously discarded channels were interpolated using spherical splines, and further time segments containing artifacts were removed using the Artifact Subspace Reconstruction method. Clean data was then saved to disk as a continuous segment and loaded into the Fieldtrip toolbox (*Oostenveld et al., 2011*) for further processing. Continuous data were segmented into 2 s or 5 s (see multiverse specifications in *Table 1*) epochs with 50% overlap. Then, segmented data were band-passed between 0.5 and 100.5 Hz for source localization. No recording was discarded based on preprocessing outcomes.

For subgroup analyses (Research Question 2), we defined two subgroups based on the clinical diagnosis: people with Chronic Back Pain (CBP, N=80) and people with Chronic Widespread Pain (CWP, N=33). To perform these analyses, we randomly drew a subsample of healthy participants of equal size for each subgroup, matched in age, gender, and dataset to which they belonged following an iterative process. An initial random sample of healthy participants containing the same number of participants per initial study as the people with chronic pain subgroup was chosen. Then differences

in age and gender between the samples were tested with Bayesian independent samples t-tests. If moderate evidence in favor of the null hypothesis was observed (BF <0.33) in both age and gender, the healthy sample was kept. If not, a different random sample of healthy participants was chosen, and the procedure was repeated.

## Aperiodic component computation

The aperiodic component is a recent measure of brain activity. Thus, how different analytical choices impact its calculation has not yet been characterized. For that reason, we performed the analyses using a two-step approach. First, we conducted the pre-registered analyses, for which we had pre-defined the steps and parameters to estimate the aperiodic activity. Second, we performed a multiverse analysis to investigate the robustness of the results across different methodological ways of computing the aperiodic activity.

Specifically, we performed a specification curve analysis (*Simonsohn et al., 2020*) in which we first identified a set of specifications, i.e., equally valid analytical options to estimate aperiodic activity. We then estimated aperiodic activity, performed the pre-defined statistical test for each specification, and visualized the results in a specification curve. We conducted three inference tests on the specification curve to assess its statistical significance (see *Statistical Analyses*). In *Table 1*, we list all the analytical decisions and their parameters, and describe their potential influence on the estimation of the aperiodic activity. The combination of all these analytical options yielded 48 different specifications, including the approach of the pre-registered analysis.

### Source localization

To spatially localize brain activity, we performed a source reconstruction of the preprocessed, segmented, and band-passed filtered signals. We projected the data to source space using an array-gain Linear Constrained Minimum Variance (LCMV) beamformer. As a source model, we used the centroid regions of the 100-parcel version of the Schaefer atlas with 17 networks (*Schaefer et al., 2018*; *Yeo et al., 2011*). The lead field was built using a realistically shaped volume conduction model based on the Montreal Neurological Institute (MNI) template available in FieldTrip (standard_bem. mat) and the source model. Spatial filters were constructed with the covariance matrices of the band-pass filtered data and the described lead fields. A 5% regularization parameter was set to account for rank deficiencies in the covariance matrix. The dipole orientation was fixed to the direction of the maximum variance following the most recent recommendations (*Westner et al., 2022*).

We defined our primary region of interest, the medial prefrontal cortex, as all the parcels from the above-mentioned version of the Schaefer atlas labeled medial prefrontal cortex ('PFCm'), medial posterior prefrontal cortex ('PFCmp'), and anterior cingulate cortex ('Cinga'). Four parcels comprised the mPFC by this definition (*Figure 1*).

### Power spectrum computation

We reconstructed the virtual time series at each source location using the spatial filter and band-pass filtered sensor-level data. Power at each source location was then calculated between 1 and 100 Hz with a Fast Fourier Transform using the Fieldtrip function ft_freqanalysis and the 'mtmfft' method. We used Slepian multitapers with +/-1 Hz frequency smoothing for the pre-registered analyses. Average power spectra values are visualized in Supp. Fig. 3. For other specifications of the multiverse analysis, we used a single Hanning taper (*Table 1*).

To investigate aperiodic activity in the mPFC, we constructed a representative power spectrum of this region by averaging the power spectra of the four source locations that defined the mPFC. This representative power spectrum of the mPFC was used later to model aperiodic activity. We investigated an alternative way to summarize mPFC activity in the multiverse analysis by estimating aperiodic activity for each source location and then averaging aperiodic parameters in the mPFC, i.e., the aperiodic exponent (*Table 1*).

### Power spectrum parametrization

Power spectra were parametrized into periodic and aperiodic components with a re-implementation in MATLAB of the 'spectparam' algorithm, formally known as FOOOF (*Donoghue et al., 2020*). The algorithm was reimplemented to integrate it with the multiverse analyses computation. The power

spectrum was modeled in the log-log scale as a sum of an aperiodic component and N oscillatory peaks, each modeled individually by a Gaussian function. The aperiodic component, L, was modeled using a Lorentzian function.

$$L\left(F\right) = b - \log\left(k + F^{\chi}\right)$$

Here, b denotes the aperiodic offset, k the 'knee' parameter, which controls for the kink in the aperiodic component, F frequency, and $\chi$ the aperiodic exponent. The power spectrum model with parameters b, k, and $\chi$ was fitted to the data in an iterative procedure as in *Donoghue et al., 2020* using non-linear least squares solvers implemented with the Matlab 'lsqcurvefit' function.

For the pre-registered analyses, we modeled the power spectrum using the settings proposed in the 'spectparam' tutorial, that is modeling the power spectrum in the 2–40 Hz range and not including the knee parameter. In the multiverse analysis, we also explored two alternative frequency ranges: the 40–60 Hz range (*Gao et al., 2017*), and the 1–100 Hz range. We also investigated the influence of including the knee parameter in estimating the aperiodic component (*Table 1*). The rest of the algorithm parameters were not modified except for the minimum detectable peak width, which was increased from 0.5 Hz to 1 Hz to match twice the frequency resolution of our data. For a detailed description of these parameters, we refer the reader to the 'spectparam' documentation (https://fooof-tools.github.io/fooof/index.html) and the pre-registration.

Additionally, we extracted two measures of goodness of fit of the modeled power spectra: the mean average error (MAE) and the explained variance ($R^2$). With the pre-registered settings, the power spectra of all participants could be satisfactorily modeled (MAE = 0.03 ± 0.01, $R^2$=0.98 ± 0.01 [mean ± std]). Thus, no participant was excluded based on a poor model fit. In the multiverse analysis, in the eight specifications in which the power spectrum was modeled between 40 and 60 Hz and the knee parameter was estimated, the fitting algorithm did not converge in a minority of participants. These participants were excluded in the statistical analysis of the respective specification. The maximum number of discarded participants in a specification was 39.

## Statistical analyses
### Effect size
In the context of pre-registration, we conducted a sensitivity analysis in G*power (*Faul et al., 2007*) to determine the effect size detectable with a two-tailed independent samples t-test. With our pre-defined sample size, given that the data had already been acquired, an α error probability of 0.05, and a statistical power of 0.95, we could detect medium effect sizes (Cohen's d=0.44).

### Statistical models
We investigated two main topics in our three research questions. First, we investigated whether the aperiodic component differs between people with chronic pain and healthy participants. Second, we explored the relationship between the aperiodic exponent and pain intensity in people with chronic pain. To control for age effects, we used linear regression models to remove the influence of age from aperiodic exponents and pain intensity ratings. Thus, all statistical tests were performed on the residuals obtained from the regression models.

Research questions 1 and 2 focused on the medial prefrontal cortex. For these research questions, we performed a Bayesian two-tailed independent samples t-test to compare the age-corrected aperiodic exponents in the mPFC between people with chronic pain and healthy participants. We also performed a Bayesian linear correlation to explore the relationship between the age-corrected aperiodic exponent in the mPFC and age-corrected pain intensity in people with chronic pain. These tests were both part of our pre-defined analyses and applied to each specification of the multiverse analysis. The results of the multiverse analysis were gathered and visualized using a specification curve, depicting each specification's effect size and Bayes Factor.

Research question 3 investigated aperiodic activity at the whole-brain level. To this end, we aimed to address the same questions as outlined above but focused on one hundred different brain locations. To our knowledge however, it is impossible to address the multiple comparison problem in a Bayesian framework. Therefore, we performed the frequentist version of the aforementioned statistical tests at each source location. Then, we adjusted the p-values with the resampling-based FDR

correction (*Yekutieli and Benjamini, 1999*). Due to the nature of the question and the high number of statistical comparisons, we decided not to perform a multiverse analysis.

We implemented the statistical tests in R (*R Development Core Team, 2021*) using the package 'BayesFactor' for the Bayesian tests and 'specr' for the visualization of the specification curve.

### Inference criteria

For research questions 1 and 2, statistical inference of the pre-defined analyses was based on Bayes Factors ($BF_{10}$) obtained from each statistical test. Moderate evidence favoring the alternative hypothesis was inferred if $BF_{10} > 3$, inconclusive evidence if $1/3 < BF_{10} < 3$, and moderate evidence favoring the null hypothesis if $BF_{10} < 1/3$. Bayes factors larger than 10 or smaller than 1/10 were considered strong evidence in favor or against the alternative hypothesis, respectively.

In the multiverse analysis, we integrated the statistical evidence from all the specifications by performing three inference tests based on a permutation approach, as proposed in the original publication (*Simonsohn et al., 2020*). Of note, these tests are based on frequentist statistics as, to the best of our knowledge, Bayesian approaches to specification curve inference do not exist yet. Thus, for each multiverse analysis, we generated a null distribution of 500 specification curves by randomly shuffling the group labels or the pain ratings and recomputing the analyses for all specifications. Then, we conducted three inference tests to assess the likelihood of obtaining the original specification curve under the null hypothesis of no effect. In each test a p-value was derived by comparing a test statistic of the original specification curve to the corresponding distribution of test statistics of the randomized specification curves. The considered test statistics were: (1) The median effect size, i.e., Cohen's d or Pearson's correlation coefficient depending on the analysis ($p_{median}$). (2) The share of specifications indicating evidence for an effect, i.e., with $BF_{10} > 3$, and the same direction of effect as the original curve ($p_{share}$). (3) Z-values associated with p-values summarized across all specifications according to Stouffer's method ($p_{aggr}$). Note that for this test a frequentist version of t-test or correlation test was computed for each specification. The inference criterion for each of the three tests was a p-value $< 0.05$.

For research question 3, we inferred statistical significance if the FDR-adjusted p-values were lower than 0.05.

### Deviations from the pre-registration

During the pre-registration of research question 1, we proposed to perform a Bayesian ANCOVA with group (people with chronic pain/healthy) as a fixed factor and age as a covariate in JASP. However, it was impossible to perform this test programmatically for all specifications during the multiverse analysis. Therefore, we switched to independent samples t-tests, available in the R package 'BayesFactor' on age-corrected aperiodic exponents.

Additionally, we pre-registered another hypothesis that depended on a positive finding concerning pre-registered research question 1. As there were no differences in aperiodic exponents in the mPFC between people with chronic pain and healthy participants, that hypothesis was not tested, and whole-brain effects were investigated for research question 3.

## Acknowledgements

The study was supported by the TUM Innovation Network Neurotechnology in Mental Health (NEUROTECH) and the Deutsche Forschungsgemeinschaft (PL 321/14–1, SFB1158).

## Additional information

### Competing interests

Markus Ploner: Reviewing editor, eLife. The other authors declare that no competing interests exist.

## Funding

| Funder | Grant reference number | Author |
|---|---|---|
| Deutsche Forschungsgemeinschaft | SFB1158 | Markus Ploner |
| Deutsche Forschungsgemeinschaft | PL321/14-1 | Markus Ploner |
| Technical University of Munich | Neurotech | Markus Ploner |

The funders had no role in study design, data collection and interpretation, or the decision to submit the work for publication.

## Author contributions

Cristina Gil Avila, Conceptualization, Data curation, Software, Formal analysis, Validation, Investigation, Visualization, Methodology, Writing – original draft, Writing – review and editing; Elisabeth S May, Conceptualization, Software, Investigation, Visualization, Writing – review and editing; Felix S Bott, Conceptualization, Software, Writing – review and editing; Laura Tiemann, Conceptualization, Investigation, Writing – review and editing; Vanessa Hohn, Writing – review and editing; Henrik Heitmann, Investigation, Writing – review and editing; Paul Theo Zebhauser, Conceptualization, Writing – review and editing; Joachim Gross, Methodology, Writing – review and editing; Markus Ploner, Conceptualization, Resources, Supervision, Funding acquisition, Visualization, Methodology, Writing – original draft, Project administration, Writing – review and editing

## Author ORCIDs

Cristina Gil Avila ⓘ https://orcid.org/0000-0003-3789-0644
Elisabeth S May ⓘ https://orcid.org/0000-0002-8558-6447
Joachim Gross ⓘ https://orcid.org/0000-0002-3994-1006
Markus Ploner ⓘ https://orcid.org/0000-0002-7767-7170

## Ethics

The study was a pre-registered analysis of public datasets.

Reviewer #1 (Public review): https://doi.org/10.7554/eLife.101727.3.sa1
Reviewer #2 (Public review): https://doi.org/10.7554/eLife.101727.3.sa2
Author response https://doi.org/10.7554/eLife.101727.3.sa3

# Additional files

## Supplementary files

MDAR checklist

## Data availability

All the EEG raw data in BIDS format (*Pernet et al., 2019*), demographic and clinical variables, and code used in this work are available in the OSF project https://osf.io/qgfma/.

The following dataset was generated:

| Author(s) | Year | Dataset title | Dataset URL | Database and Identifier |
|---|---|---|---|---|
| Gil Avila C, Ploner M | 2023 | 1/f in chronic pain | https://osf.io/qgfma/ | Open Science Framework, qgfma |

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
