## [Editor Report · eLife Assessment]

Gil Ávila et al. evaluated the aperiodic component in the medial prefrontal cortex using resting-state EEG recordings from 149 individuals with chronic pain and 115 healthy participants. The authors present **compelling** evidence that the aperiodic component of the EEG does not differentiate between those with chronic pain and healthy individuals. The study was well-designed and rigorously conducted, and the clear and conclusive results provide **important** insights that can guide future research in the field of pain neuroscience.

---

## [Referee Report · Reviewer #1 (Public review)]

Summary:

In this study, Avila et al. tested the hypothesis that chronic pain states are associated with changes in excitability of the medial prefrontal cortex (mPFC). The authors used the slope of the aperiodic component of the EEG power spectrum = the aperiodic exponent as a novel, non-invasive proxy for the cortical excitation-inhibition ratio. They performed source localization to estimate the EEG signals generated specifically by the mPFC. By pooling resting-state EEG recordings from three existing datasets, the authors were able to compare the aperiodic exponent in the mPFC and across the whole brain (at all modeled cortical sources) between 149 chronic pain patients and 115 healthy controls. Additionally, they assessed the relationship between the aperiodic exponent and pain intensity reported by the patients. To account for heterogeneity in pain etiology, the analysis was also performed separately for two patient subgroups with different chronic pain conditions (chronic back pain and chronic widespread pain). The study found robust evidence against differences in the aperiodic exponent in the mPFC between people with chronic pain and healthy participants, and no correlation was observed between the aperiodic exponent and pain intensity. These findings were consistent across different patient subgroups and were corroborated by the whole-brain analysis.

Strengths:

The study is based on sound scientific reasoning and rigorously employs suitable methods to test the hypothesis. It follows a pre-registered protocol, which greatly increases the transparency and, consequently, the credibility of the reported results. In addition to the planned steps, the authors used a multiverse analysis to ensure the robustness of the results across different methodological choices. I find this particularly interesting, as the EEG aperiodic exponent has only recently been linked to network excitability, and the most appropriate methods for its extraction and analysis are still being determined. The methods are clearly and comprehensively described, making this paper very useful for researchers planning similar studies. The results are convincing, supported by informative figures, and the lack of the expected difference in mPFC excitability between the tested groups is thoroughly and constructively discussed.

Weaknesses:

Firstly, to augment the sample size, the authors pooled data recorded by different researchers using different experimental protocols. This inevitably increases sample variability and may limit the availability of certain measures, as was the case here with the reports of pain intensity in the patient group. Secondly, the analysis heavily relies on the estimation of cortical sources, an approach that may yield imprecise results, especially when default conduction models, source models, and electrode coordinates are used (as was the case here).

Comments on revisions:

The authors satisfactorily revised the manuscript and responded to previous questions and suggestions. I have no further comments.

---

## [Referee Report · Reviewer #2 (Public review)]

Summary:

This study evaluated the aperiodic component in the medial prefrontal cortex (mPFC) using resting-state EEG recordings from 149 individuals with chronic pain and 115 healthy participants. The findings showed no significant differences in the aperiodic component of the mPFC between the two groups, nor was there any correlation between the aperiodic component and pain intensity. These results were consistent across various chronic pain subtypes and were corroborated by whole-brain analyses. The study's robustness was further reinforced by preregistration and multiverse analyses, which accounted for a wide range of methodological choices.

Strengths:

This study was rigorously conducted, yielding clear and conclusive results. Furthermore, it adhered to stringent open and reproducible science practices, including preregistration, blinded data analysis, and Bayesian hypothesis testing. All data and code have been made openly available, underscoring the study's commitment to transparency and reproducibility.

Weaknesses:

The aperiodic exponent of the EEG power spectrum is often regarded as an indicator of the excitatory/inhibitory (E/I) balance. However, this measure may not be the most accurate or optimal for quantifying E/I balance, a limitation that the authors might consider addressing in the future.

Comments on revisions:

All my comments have been well addressed.

---

## [Author Response]

The following is the authors’ response to the original reviews.

**Public Reviews**

**Reviewer #1:**
Summary:In this study, Avila et al. tested the hypothesis that chronic pain states are associated with changes in the excitability of the medial prefrontal cortex (mPFC). The authors used the slope of the aperiodic component of the EEG power spectrum = the aperiodic exponent as a novel, non-invasive proxy for the cortical excitation-inhibition ratio. They performed source localization to estimate the EEG signals generated specifically by the mPFC. By pooling resting-state EEG recordings from three existing datasets, the authors were able to compare the aperiodic exponent in the mPFC and across the whole brain (at all modeled cortical sources) between 149 chronic pain patients and 115 healthy controls. Additionally, they assessed the relationship between the aperiodic exponent and pain intensity reported by the patients. To account for heterogeneity in pain etiology, the analysis was also performed separately for two patient subgroups with different chronic pain conditions (chronic back pain and chronic widespread pain). The study found robust evidence against differences in the aperiodic exponent in the mPFC between people with chronic pain and healthy participants, and no correlation was observed between the aperiodic exponent and pain intensity. These findings were consistent across different patient subgroups and were corroborated by the whole-brain analysis.Strengths:The study is based on sound scientific reasoning and rigorously employs suitable methods to test the hypothesis. It follows a pre-registered protocol, which greatly increases the transparency and, consequently, the credibility of the reported results. In addition to the planned steps, the authors used a multiverse analysis to ensure the robustness of the results across different methodological choices. I find this particularly interesting, as the EEG aperiodic exponent has only recently been linked to network excitability, and the most appropriate methods for its extraction and analysis are still being determined. The methods are clearly and comprehensively described, making this paper very useful for researchers planning similar studies. The results are convincing, and supported by informative figures, and the lack of the expected difference in mPFC excitability between the tested groups is thoroughly and constructively discussed.

We are grateful for the appreciation of the strengths of our study.

Weaknesses:Firstly, although I appreciate the relatively large sample size, pooling data recorded by different researchers using different experimental protocols inevitably increases sample variability and may limit the availability of certain measures, as was the case here with the reports of pain intensity in the patient group. Secondly, the analysis heavily relies on the estimation of cortical sources, an approach that offers many advantages but may yield imprecise results, especially when default conduction models, source models, and electrode coordinates are used. In my opinion, this point should be discussed as well.

We agree that the heterogeneous sample of people with chronic pain increases variability and limits the availability of clinical measures. We further agree on the limitations of source space analysis. Therefore, we have added these limitations to the discussion section.

**Reviewer #2:**
Summary:This study evaluated the aperiodic component in the medial prefrontal cortex (mPFC) using restingstate EEG recordings from 149 individuals with chronic pain and 115 healthy participants. The findings showed no significant differences in the aperiodic component of the mPFC between the two groups, nor was there any correlation between the aperiodic component and pain intensity. These results were consistent across various chronic pain subtypes and were corroborated by whole-brain analyses. The study's robustness was further reinforced by preregistration and multiverse analyses, which accounted for a wide range of methodological choices.Strengths:This study was rigorously conducted, yielding clear and conclusive results. Furthermore, it adhered to stringent open and reproducible science practices, including preregistration, blinded data analysis, and Bayesian hypothesis testing. All data and code have been made openly available, underscoring the study's commitment to transparency and reproducibility.

We appreciate the appraisal of the strengths of our study, highlighting our efforts in open and reproducible science practices.

Weaknesses:The aperiodic exponent of the EEG power spectrum is often regarded as an indicator of the excitatory/inhibitory (E/I) balance. However, this measure may not be the most accurate or optimal for quantifying E/I balance, a limitation that the authors might consider addressing in the future.

We are grateful for this suggestion and fully agree that the aperiodic component of the power spectrum is not necessarily the most optimal and accurate measure for quantifying E/I balance. We have now included this limitation in the discussion section.

**Recommendations for the authors**

**Reviewer #1:**
(1) In the Results section, it might be helpful to provide the mean values of the aperiodic exponent (before age correction) for all tested groups and subgroups. As this measure is still not widely used, providing these values would allow readers to better understand the normal range of the aperiodic exponent.

We have added the mean values of the aperiodic exponent and their standard deviation (before age correction) to the manuscript's results section (page 6 and 11).

(2) When reporting the aperiodic exponent across all cortical sources (Q3), I think it would be useful to include the raw values in Figure 6 in the main text rather than in the Supplementary Materials. At a glance, these plots seem to suggest that the aperiodic exponent differs between groups in the occipital and parietal regions, even though no tests were significant after correcting for multiple comparisons. Maybe this observation also deserves a mention in the text and possibly in the Discussion..?

We have moved the report on the aperiodic exponent across all cortical sources from the Supplementary Material to the main text. It is now Fig. 7 of the main manuscript. Moreover, we agree that the plots suggest group differences in certain brain regions. However, according to our rigorous open and reproducible science practices and pre-registration, we prefer not to speculate on these non-significant findings.

(3) In the Methods section, when describing the participants, the authors state that "Gender was balanced across both groups...". It might be better to avoid referring to the datasets as "balanced," considering that the sample includes almost twice as many females as males.

We have replaced the misleading statement with the more precise statement that ”the gender ratio of both groups was similar.”

(4) In the Methods section, when describing the source localization, I find it slightly confusing that the authors first mention the anterior cingulate cortex as a possible label included in the mPFC cortical parcels but then state that the version of the cortical atlas used did not contain such a label. It might be simpler not to mention the cingulate cortex at all.

We have deleted the misleading sentence from the manuscript.

**Reviewer #2:**
(1) The aperiodic exponent of the EEG power spectrum is often considered an indicator of the excitatory/inhibitory (E/I) balance, but this measure can be susceptible to artifacts. It is important to acknowledge this limitation and consider exploring alternative measures to quantify the E/I ratio in future studies.

We are grateful for this suggestion and fully agree that the aperiodic component of the power spectrum is not necessarily the most optimal and accurate measure for quantifying E/I balance. We have now included this limitation in the discussion section.

(2) The study assumed a linear relationship between the E/I ratio (represented by the aperiodic exponent of the EEG power spectrum) and chronic pain. However, this assumption may not hold true in all cases, and this point could be discussed in the study.

We fully agree that the relationship between the E/I ratio and chronic pain might not be a linear one and have added this point to the discussion section.

(3) The aperiodic component was characterized in eyes-closed resting-state EEG recordings, although EEG data were collected in both eyes-closed and eyes-open conditions. The authors could also consider assessing the aperiodic component from EEG data with eyes open.

We thank the reviewer for this suggestion. We have focused our analysis on eyes-closed recordings since these recordings are usually less contaminated by artifacts than eyes-open recordings. Moreover, in our current datasets, some participants were missing eyes-open recordings. We agree that performing similar analyses for the eyes-open recordings would also be interesting. However, adding these analyses would double the amount of data included in the manuscript, which would likely overload it. We have, therefore, now included a statement to the discussion that future studies should also analyze eyes-open EEG recordings.

(4) The EEG power spectrum was calculated from signals after source reconstruction, a crucial step for targeting specific brain regions. However, this process can introduce potential signal distortions, such as variations in source waveforms depending on different regularization parameters. To ensure the robustness of the results, the authors could perform the same analysis at the sensor level, for example, using signals recorded at Fz.

We agree on the potential shortcomings and limitations of source space analysis and have added this limitation to the discussion section.

(5) It would be beneficial to present the raw EEG power spectrum averaged across subjects for each condition, along with the scalp distribution of the aperiodic exponent. This would enhance readers' understanding of the study and help demonstrate the quality of the data.

We are grateful for this suggestion and added the power spectrum for each condition and the scalp distribution of the aperiodic exponent to the Supplementary Material.

(6) Linear regression models were used to control for the influence of age on aperiodic exponents and pain intensity ratings. However, it is unclear why other relevant variables, such as gender and medication use, were not considered.

We agree that the aperiodic exponent might be influenced by gender and medication. As these analyses had not been included in our pre-registered analysis plan, we have not performed them. Moreover, although we agree that gender might have an impact, we have not found any evidence for this so far. Regarding medication, we fully agree that medication can influence the measure. However, medication was very heterogeneous, including drugs with fundamentally different mechanisms of action. Thus, we do not see a robust way to appropriately analyze these effects with sufficient statistical power. We have now added this important point to the discussion section.

(7) The authors may consider addressing or discussing the impact of inter-individual variability on the negative results, particularly given that the data were derived from multiple experiments.

We agree that the heterogeneous sample of people with chronic pain increases variability and limits the availability of clinical measures. We have added this limitation to the discussion.